# Refractive Index Fiber Laser Sensor by Using a Fiber Ball Lens Interferometer with Adjustable Free Spectral Range

**DOI:** 10.3390/s23063045

**Published:** 2023-03-11

**Authors:** Ricardo Iván Álvarez-Tamayo, Patricia Prieto-Cortés

**Affiliations:** 1Faculty of Mechatronics, Bionics and Aerospace, Universidad Popular Autónoma del Estado de Puebla, Puebla 72410, Mexico; 2Mechatronics Division, Universidad Tecnológica de Puebla, Puebla 72300, Mexico

**Keywords:** optical fiber, laser sensors, fiber ball lens, interferometric spectral filters, refractometers, EDF fiber lasers

## Abstract

In this work, a fiber laser refractometer based on a fiber ball lens (FBL) interferometer is proposed. The linear cavity erbium-doped fiber laser uses an FBL structure acting as a spectral filter and sensing element for determining the RI of a liquid medium surrounding the fiber. The optical interrogation of the sensor is the wavelength displacement of the generated laser line as a function of the RI variations. For the proposed FBL interferometric filter, the free spectral range of its wavelength-modulated reflection spectrum is adjusted to maximum in order to obtain RI measurements in a range of 1.3939 to 1.4237 RIU, from laser wavelength displacements in a range from 1532.72 to 1565.76 nm. The obtained results show that the wavelength of the generated laser line is a linear function of the RI variations on the medium surrounding the FBL with a sensitivity of 1130.28 nm/RIU. The reliability of the proposed fiber laser RI sensor is analytically and experimentally investigated.

## 1. Introduction

Fiber refractometers have been of persistent interest for sensing applications in chemical, biological, medical instrumentation systems, and marine environment monitoring [1,2,3,4,5] because of their intrinsic advantages such as compactness, high sensitivity, and low-cost implementation. Among the various fiber optic refractometers, those based on the use of interferometric filters have proven their reliability due to their ease of implementation, variety of interference methods, and flexibility in the setting of operating parameters [6,7,8]. For this purpose, the use of micro-machined special fibers has allowed the implementation of robust, compact, stable, and flexible interferometers suitable to be integrated to more complex optical systems. Moreover, the use of fiber ball lenses (FBL) has been proven for the development of functional optical resonators satisfying the principle of minimum energy [9], and as high efficiency light couplers between fibers [10,11]. Recently, Jasim et al. [12] proposed the implementation of an interferometer based on the use of an FBL and a reflecting mirror. The optical setup acts as a two-path multi-beam interferometer due to a couple of reflective surfaces generating a wavelength-modulated reflection output spectrum. Based on the FBL interferometer (FBLI), broadband fiber sensors have been reported for the measurement of different variables such as displacement [12], glucose in ionized water [13], humidity [14,15], and formaldehyde [16,17], with improved and repeatable results. For these reported approaches, the interrogation method is the determination of the free spectral range (FSR) variation as a function of the measured variable. Then, the adjustable FSR can be an attractive characteristic of the FBLI for its use as a spectral filter in different applications including tunable and multi-wavelength fiber lasers [18,19] and, potentially, for optical sensors with different interrogation methods.

Moreover, laser sensors have proven the improvement of wavelength interrogation optical sensors compared with broadband-spectrum sensors due to their well-defined narrow band spectrum which enables the high precision determination of the generated laser line [20,21,22]. Here, the straightforward determination of the wavelength displacement of a narrow-width laser spectrum is precisely determined, unlike traditional broadband spectrum sensors where the wavelength at which the maximum intensity is obtained is not easy, leading to measurement errors. In addition, the quality of the sensor is significantly improved as it can be estimated by the Q value [22]. In this regard, the use of interferometers and other wavelength tunable optical filters along with a fiber laser cavity represent a suitable configuration to develop high precision fiber sensors [23,24,25,26]. However, the use of the FBLI has been unexplored for the development of fiber laser refractometers. In this case, the use of the FBLI structure represents a feasible alternative due to its advantages of ease of construction and flexibility in adjusting the operation parameters [12,13].

In this paper, for the first time, to the best of our knowledge, we propose a fiber laser refractometer by using an FBLI interferometer as a spectral filter. By the wavelength shift of the laser line, the linear cavity erbium-doped fiber (EDF) laser uses the FBL as a sensing element for the determination of the RI of a liquid medium surrounding the fiber structure. For the used FBLI, the FSR of its reflection spectrum is adjusted to obtain the RI estimation in a broad laser emission wavelength range. The proposed sensor using an FBL interferometer within a laser cavity significantly improves the refractive index detection compared with the reported sensors using the FBL interferometer for detection directly from the interference spectrum. The determination of the wavelength displacement as the optical interrogation is more precise and straightforward to determine from a laser line than from a broadband spectrum.

## 2. Fiber Ball Lens Interferometer

The FBL consist of a quasi-spherical-shaped tip attached to the fiber body. Typically, the fiber ball is fabricated by a controlled electric arc discharge on the fiber tip, from the manual operation of a fusion splicer. The discharge parameters such as time and power are optimized to control the ball diameter and shape, from both single-mode and multimode fibers [27,28,29,30]. Nowadays, fusion splicers with features for special fibers fabrication, such as fiber ball lenses and tapered fibers, are currently available. In our case, instead of manual fabrication, the FBL used in our approach was fabricated from a multimode fiber with a core diameter of 50 µm (Thorlabs FG050LGA, Newton, NJ, USA), by using the dedicated ball lens arc fusion program (AFL Fiber Processing Software FPS, ver. 1.2b, Tokyo, Japan) of a fusion splicer with special fiber processing features (Fujikura ArcMaster FSM-100M, Tokyo, Japan), where the fabrication process is controlled and monitored with high precision ensuring the consistency and repeatability of the FBL fabrication. Based on the performance results obtained by Li et al. [27] for the fabrication of MMF-based FBL structures, the ball diameter was adjusted to 300 µm with minimal ellipticity. The results obtained in ref. [27] shows that large-diameter FBLs are recommended as follow-up experimental optical devices. The microscope image of the fabricated FBL is shown in Figure 1.

The FBL interferometer (FBLI) is based on the multiple-beam interference configuration proposed by A.A. Jasim et al. [12], shown in Figure 2. A reflecting mirror is placed in front of the FBL cavity at an adjustable distance d. The input light $I_{in}$ from the fiber core is propagated in multiple modes within the FBL cavity. Then, two different interfaces act as reflecting surfaces: the interface formed by the FBL surface with the free space, and the one formed between the free space and the mirror. A portion of the input light with intensity $I_{1}$ meet the conditions to be reflected by the FBL inner surface back to the fiber. On the other hand, another part of the light with intensity $I_{2}$ surpasses the FBL surface, propagates through the space with length d, is reflected by the mirror, and is coupled back to the fiber using the FBL as a light coupler.

The reflected beams from $I_{1}$ and $I_{2}$ interfere back to the fiber with a phase difference due to the optical path difference between two beams, given by [12]
(1)$$\Delta \phi =\frac{4\pi }{\lambda }n_{g}d+\phi_{0},$$
where λ is the operating wavelength and $n_{g}$ is the refractive index (RI) of the medium surrounding the gap between the FBL and the mirror.

The light propagation through the FBLI was simulated by using the Optiwave OptiFDTD software. The parameters of the fabricated FBL were used. A fiber ball with a diameter of 300 µm and an MMF with a core diameter of 50 µm and cladding diameter of 125 µm were simulated. The input source was normalized with maximal intensity at 1550 nm. The main objective of the beam trace simulation is to confirm the experimental results obtained by Jasim et al. [11] for the microscope image describing the FBL-reflected light from its inner surface. Then, the objective is to investigate the second reflected intensity from the reflector mirror in order to obtain the reflection interference back to the fiber. Figure 3a shows the xz plane distribution of the light propagation through the optical axis. The input light propagates as different modes through the MMF core; then, when the fiber ball is reached, it acts as coupling element. As a result, most of the light intensity from the multiple modes remain their propagation at the central area of the XY plane, without significant dispersion within the ball. Part of the light is reflected at the inner surface of the FBL whereas a portion of light passes the FBL’s surface being propagated through the free space, reaching the mirror where it is reflected and coupled back to the FBL. As a result, the interference between the two parts of light reflected with different optical paths is observed. An amplitude maximum is observed near to the central point of the fiber ball according to the FBL cavity length, also shown in the 3D image of the light propagation through the FBLI in Figure 3b. The multiple beam interference is observed all over the FBL surface at the XZ plane, with a high amplitude at the x axis center region. As it can be observed, for the FBL dimensions and the reflector mirror distance, the focusing point where the maximal intensity of the interfered light is near to the center of the FBL is at ~400 nm of the light propagation through the optical axis. The quality of the interfered beam inside the FBL is investigated in Figure 3c where the two-dimensional light intensity distribution (XY plane) cut at 390 µm is shown. As it can be observed, the light intensity distribution exhibits wavelength modulation due to the interference between the two reflecting surfaces with an HWHM of 132.8 µm. The results confirm that the FBL is acting as a coupling element as light is propagated back and forth, where reflected beams interfere back to the fiber.

The FSR of the interference periodical wavelength modulation can be expressed as [18]:(2)$$\Delta \lambda =\frac{\lambda^{2}}{2d}$$

The numerical simulation for a two-beam optical interference model, with the phase difference described in Equation (1), is shown in Figure 4. As it can be observed from Figure 4a and described in Equation (2), the FSR of the FBLI periodical-reflected output spectrum can be varied by the fine adjustment of the distance d. The FSR of the interference spectrum decreases with the increase of the gap. For the simulated results, a high FSR variation of 31.6 nm for d changes in a range from 20 to 80 µm is noticed. Moreover, Figure 4b shows the behavior of the FBLI output spectrum for variations on the RI of the gap between the FBL and the reflector mirror. As a result, the wavelength displacement of the interference spectrum towards longer wavelengths as a linear function of the RI of the gap increase is observed. The wavelength of the interference spectrum shifts from ~22 nm for RI variations in a range from 1.4 to 1.42 RIU.

## 3. Experimental Setup

The experimental setup of the proposed fiber laser refractometer based on the use of a fiber ball lens is shown in Figure 5a. As the gain medium, the linear cavity laser uses a 2.2 m long single-mode erbium-doped fiber (EDF) (CorActive, EDF-L 1500) with a core absorption of 21 dB/m at 1530 nm, cladding absorption of 12 dB/m at 980 nm, and numerical aperture of 0.25. The EDF is pumped through a 980/1550 nm wavelength division multiplexer (WMD) by a 980 nm laser diode (LD) with power of 200 mW. At one end, the cavity is limited by a 90/10 optical coupler with interconnected output ports forming a 64% transmission fiber loop mirror (FLM). At the other end, the cavity is limited by an FBLI, formed by the FBL along with a flat NIR broadband dielectric mirror (Thorlabs, BB03-E04). The FBLI acts as a spectral filter for the laser generation and as a sensing element in contact with the liquid filling the gap. As it can be expected, the wavelength-modulated spectrum of the FBLI selects the laser emission at the wavelength for the maximum amplitude. In addition, the generated laser line is wavelength tuned as a function of the refractive index variations on the medium surrounding the FBLI fulfilling the gap. Figure 5b shows a schematic detail of the FBLI. By using a V-Groove fiber holder, the FBL was attached and fixed to an XYZ micrometric translation stage, perpendicularly aligned in front of the reflector mirror surface. The mirror was placed on the bottom face of a container into which the sensed liquid is poured. Then, by micrometrical vertical displacement on the FBL, the distance between the FBL and the reflector mirror is varied to adjust the free spectral range (FSR) of the interference spectrum. Once the distance is adjusted, the container is fulfilled with the liquid, covering the FBL, to perform the refractive index measurement of the FBL-surrounding medium. The unconnected port of the 90/10 optical coupler is the laser output, where an optical spectrum analyzer (OSA) with a maximum resolution of 0.03 nm is used to measure the laser wavelength displacement as the optical interrogation for the liquid refractive index variations.

## 4. Results and Discussion

With the purpose of optimizing the FBLI for its use as a spectral filter for laser emission, the reflected output signal of the modulated spectrum was characterized. Thus, the FLM was disconnected from the experimental setup (between the coupler port and the WDM) to open the cavity. At the unconnected port of the WDM, the reflected spectrum of the FBLI due to the EDF amplified spontaneous emission (ASE) was measured by using an OSA.

For the design of laser sensors based on the use of interferometric filters for interrogation by displacement of the laser wavelength, it is suitable to set a long FSR of the modulated spectrum that allows for obtaining a greater operating range of the sensor, limited by the dual laser emission. In this regard, by means of micrometric displacement in the translation stage, the distance d was reduced to a minimum in order to obtain the maximum FSR of the FBL filter. Figure 6 shows the experimental results of the FBLI reflected coefficient for liquid refractive index variations by using the EDF ASE as the input signal. The FSR of the wavelength-modulated spectrum is ~32 nm. The laser emission is expected to be generated at the reflection minimum within the ASE wavelength range. The reflection signal was normalized as the reflection coefficient from the measured EDF ASE spectrum shown in the inset of Figure 6. The insertion losses of the interferometric filter are of ~55%. A set of characterized glycerol solutions was used to vary the RI of the medium surrounding the FBL. For the spectrum characterization, the RI was varied from 1.3939 to 1.4095 RIU. As it can be observed, the increase on the RI of the gap wavelength displaces the reflected periodical spectrum toward longer wavelengths.

The operation of the fiber laser refractometer is shown in Figure 7. A set of eight RI variations were carried out in a range from 1.3939 to 1.4237 RIU. Figure 7a shows the laser spectra measured at the laser output port with an OSA. As it can be observed, the laser line is wavelength displaced towards longer wavelengths as the refractive index of the FBL-surrounding medium is increased. For the lower RI value of 1.3939, the laser emission exhibits dual wavelength laser generation with a higher power at the shorter wavelength laser line at ~1530.3 nm. Conversely, for the higher RI value of 1.4237, the shorter wavelength laser line of the dual wavelength laser generation starts to appear. Then, the lowest and highest RI values, for which the dual emission with equal powers would be obtained, set the operating limits of the refractometer. This operating range is in agreement with the FSR of the FBLI. Figure 7b depicts the central wavelength of the generated laser line as a function of the refractive index variations. As it can be observed, the wavelength at which the laser line is obtained acts as the high precision optical interrogation for the RI variations. The wavelength displacement of the laser line as a function of the RI changes on the medium surrounding the FBL can be linearly fitted with a slope of 1130.28 nm/RIU. The residual wavelength error is also shown for each RI variation compared with the linear fit value. The worst-case error observed over the RI testing range is of 0.93 nm, obtained for the RI value of 1.3976. The calculated mean absolute error is 0.292 nm.

In order to characterize the repeatability of the measured laser central wavelength as a function of the RI variations, a set of 15 sequential measurements from the lower to the higher RI were carried out. Then, the calculated mean absolute error (MAE) from the average wavelength for each RI measurement was obtained. From the obtained results, the repeatability of the laser line central wavelength for the RI variations is shown in Figure 8. The MAE values are shown at the corresponding error bar for each tested RI. As it can be observed, the worst-case wavelength error value observed was ±0.0371 nm corresponding to a liquid RI of 1.3976 nm. The calculated MAEs for the 15 RI index scans are around the OSA resolution; as a result, it can be inferred that, due to the wavelength interrogation method, the resolution of the proposed fiber laser sensor is limited to the OSA wavelength resolution.

Figure 9 shows the experimental results on the laser line stability of the sensor. For two different RI levels, the measurements grouped in a set of 15 laser spectra captured every 5 min were obtained to discuss the long-term stability of the sensor. Figure 9a shows the obtained captures of the generated laser line spectra for a liquid medium surrounding the fiber structure with an RI of 1.4009, whereas Figure 9b shows the spectra for laser lines generated with an RI of 1.4095. The obtained results are shown in linear scale. Good stability of the output power and the wavelength for each set of measurements is observed. An in-depth analysis for the output power and wavelength stability of the generated laser line for each RI level of 1.4009 and 1.4095 is shown in Figure 9c and Figure 9d, respectively. The output peak power and the central wavelength of the captured laser line spectra are depicted for each RI level. For the RI of 1.4009, the average power of 118.1 mW exhibits variations within the 4.7%, whereas for the RI of 1.4095, the peak power variations are within 6.7% from an average power of 120.8 mW. However, due to the interrogation method of the sensor, the most important indicator lies in the stability of the wavelength of the generated laser line. Thus, for the RI of 1.4009, the wavelength variation is in a range of 0.46 nm with an average wavelength of 1540.63 nm, which is 0.024 nm away from the theoretical value obtained by the linear fit. For the RI of 1.4095, the wavelength variations are in a range of 0.41 nm in an average central wavelength of the laser line of 1550.32 nm, which is 0.021 nm away from the theoretical value.

It is worth noting that the sensitivity of the FBL to temperature variations was recently studied by Zhang et al. in ref. [21]. For the reported fiber sensor, the crosstalk temperature of around 40 pm/°C compromised the liquid level measurement. In our case, we assume that the low sensitivity to temperature of the FBL structure can be neglected since a high sensitivity to the refractive index of 1130.28 nm/RIU was obtained for the proposed fiber laser refractometer.

In the case of optical sensors by interrogation by wavelength displacement, the main advantage of laser sensors over broad-spectrum ones lies fundamentally in the ability to determine with greater precision the wavelength to which the spectrum is shifted, due to the narrow bandwidth of the laser emission spectrum. This can be quantified through the measurement of the Q value which describes the quality of the sensor in terms of the sensing parameters and the spectral bandwidth of the optical spectrum, as follows [22]:(3)$$Q=\frac{{\mathrm{KVS}}^{2}}{\mathrm{FWHM}}$$

Q is a dimensionless factor where S is the sensitivity of the sensor and V is the spectrum visibility. K is a normalizing unit coefficient. As it can be noticed, a laser sensor significantly improves the Q value since the FWHM and the visibility are highly decreased and increased, respectively, compared with broadband spectrum optical sensors. For the proposed fiber laser refractometer, the generated laser line exhibits a visibility of ~45.2 dB and a FWHM of ~0.018 nm. Then, the estimated quality factor Q is ~3.2 × 10^9^. Then, in order to compare the reliability of the proposed fiber sensor, the obtained results are compared with reported fiber-based refractometers operating in the 1.55 µm, in terms of the quality factor Q, as it is shown in Table 1. As it can be observed, the Q value obtained from the fiber laser refractometers is highly increased compared to the broadband spectrum sensors due to the narrow FWHM and the high visibility of the laser line. Furthermore, the fiber structure used determines the sensitivity of the sensor. In this regard, plasmonic sensors based on material coated no-core fibers (NCF) exhibit high sensitivity but at the cost of a broadened optical spectra compromising the wavelength shift measurement. In the case of ref. [23], a high sensitivity of a fiber sensor is reported; however, the fiber structure is not easy to construct, compromising the reproducibility of the sensor. Additionally, the narrow FSR limits the measurement to a short refractive index range.

## 5. Conclusions

In this paper, a fiber laser refractive index sensor based on the use of an FBLI was demonstrated. The fiber refractometer estimates the RI of the liquid medium surrounding the FBL structure by wavelength shift interrogation of the generated laser line. With a wavelength-modulated of the FBLI with an FSR of ~32 nm, estimations of the RI in a range from 1.3939 to 1.4237 RIU were obtained by laser central wavelength shift in a range from 1532.72 to 1565.76 nm. The laser wavelength exhibits a linear dependence to RI variations with a sensitivity of 1130.28 nm/RIU and a quality factor Q of ~3.21 × 10^9^. The reliability of the proposed fiber laser refractometer with advantages of ease of implementation and interrogation, flexibility, robustness, and accuracy were experimentally demonstrated.

## Figures and Tables

**Figure 1 sensors-23-03045-f001:**
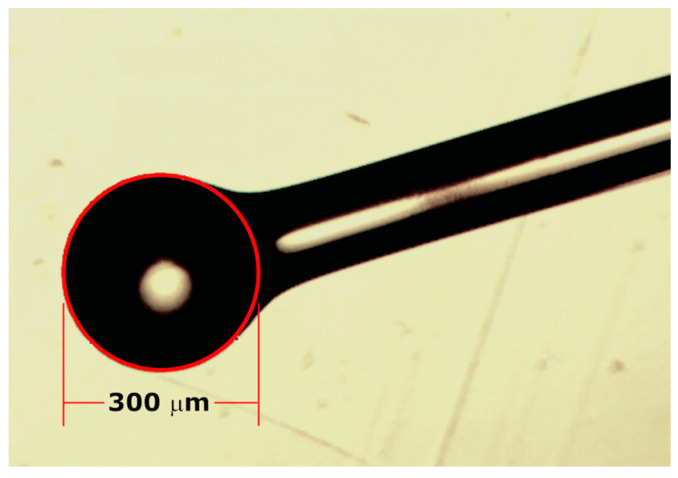
Microscope image of the fabricated multimode FBL with a ball diameter of 300 µm.

**Figure 2 sensors-23-03045-f002:**
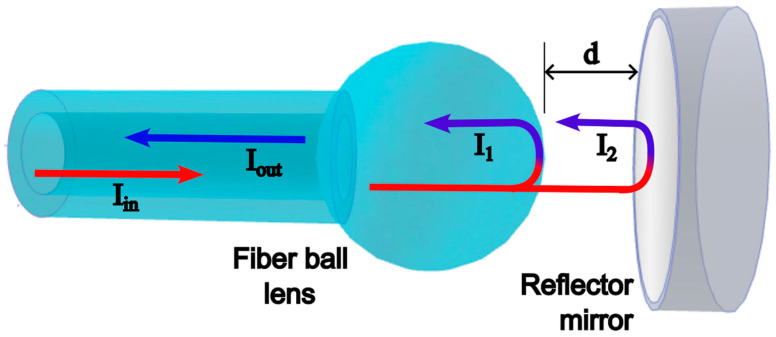
Schematic of the multiple-beam FBLI.

**Figure 3 sensors-23-03045-f003:**
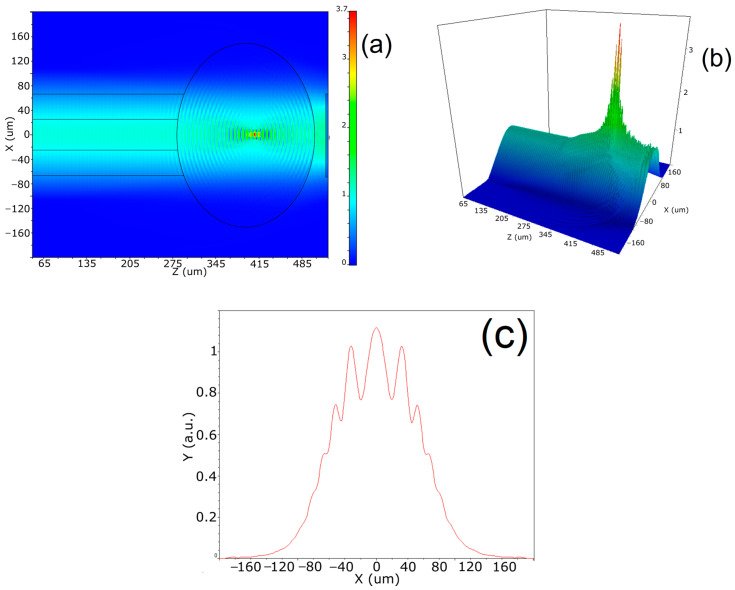
Beam trace simulation of intensity distribution for the FBLI: (**a**) XZ plane intensity distribution; (**b**) 3D intensity distribution; (**c**) XY plane cut at 390 µm of z length.

**Figure 4 sensors-23-03045-f004:**
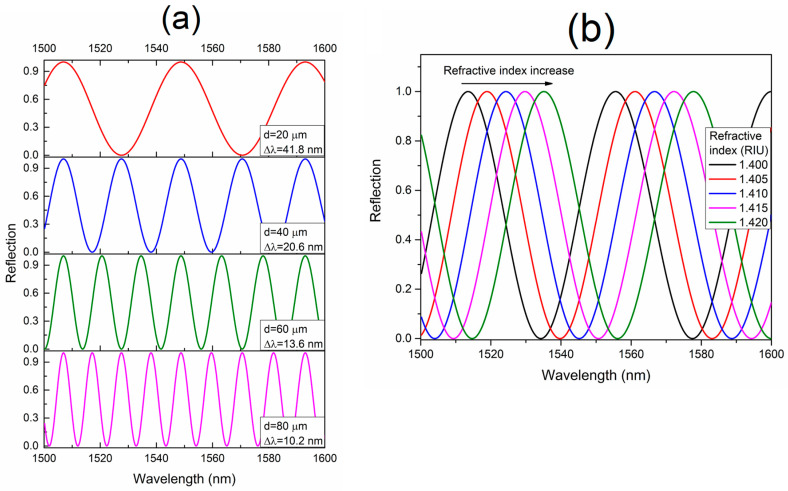
Numerical simulation of the FBL interference reflected spectrum (**a**,**b**).

**Figure 5 sensors-23-03045-f005:**
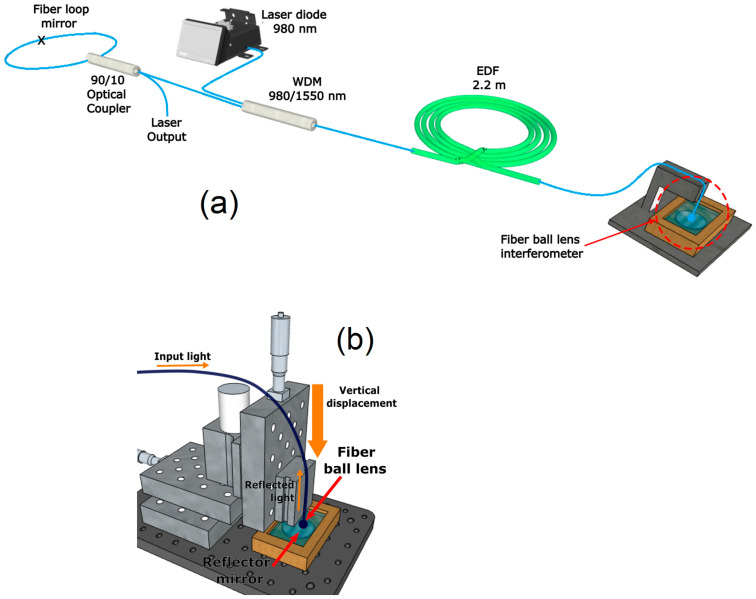
Experimental setup of the FBL interference fiber laser refractometer: (**a**) Configuration of the linear cavity laser configuration; and (**b**) detailed schematic of the FBLI implementation.

**Figure 6 sensors-23-03045-f006:**
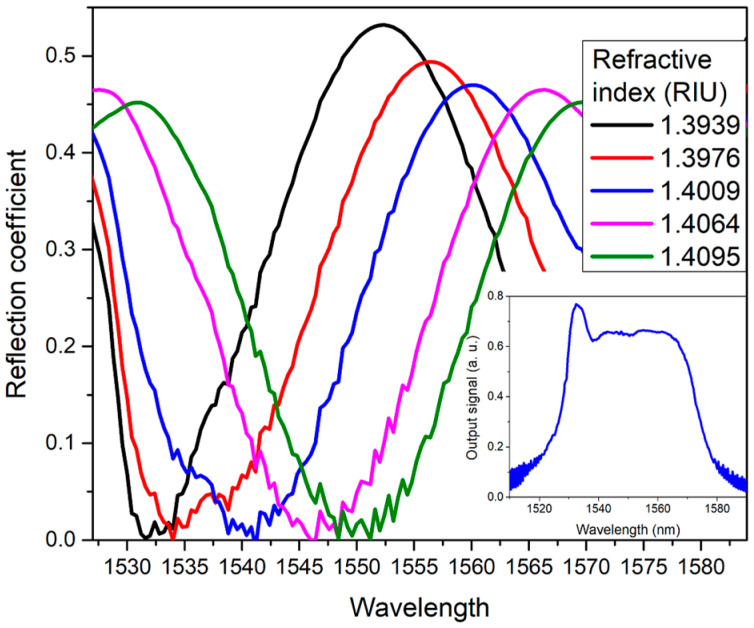
Reflection coefficient of the FBLI wavelength-modulated spectra for variations of RI on the gap between the FBL and the reflector mirror with EDF ASE as the input signal. Inset: Measured spectrum of the EDF ASE.

**Figure 7 sensors-23-03045-f007:**
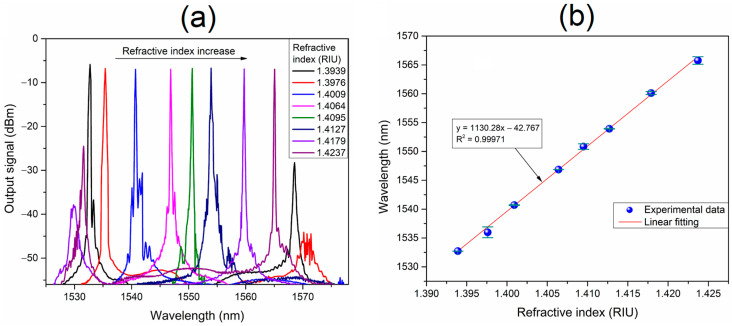
Operation of the FBL-based laser refractometer: (**a**) the output signal of the laser sensor for RI variations on the medium surrounding the FBL; and (**b**) laser central wavelength displacement as a function of the RI changes.

**Figure 8 sensors-23-03045-f008:**
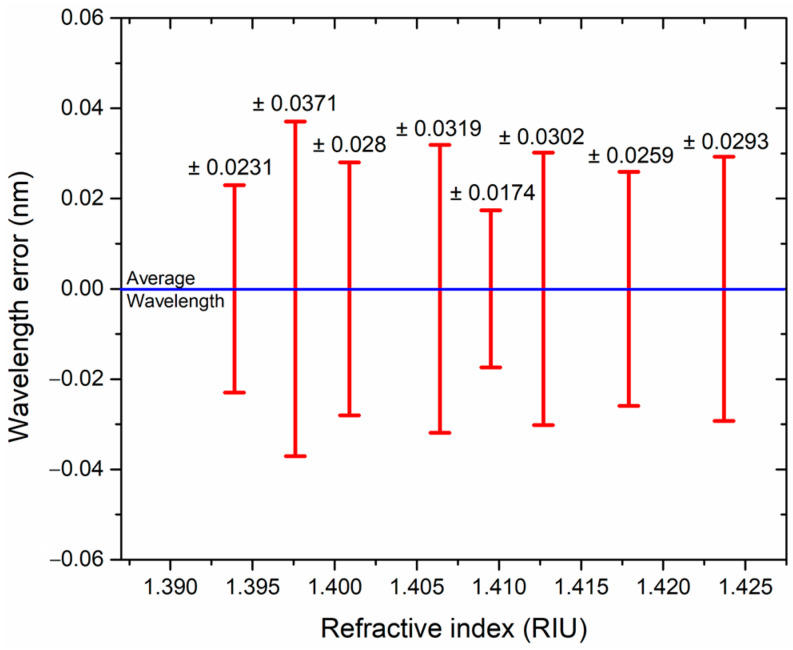
Repeatability of wavelength displacement for the RI fiber laser sensor.

**Figure 9 sensors-23-03045-f009:**
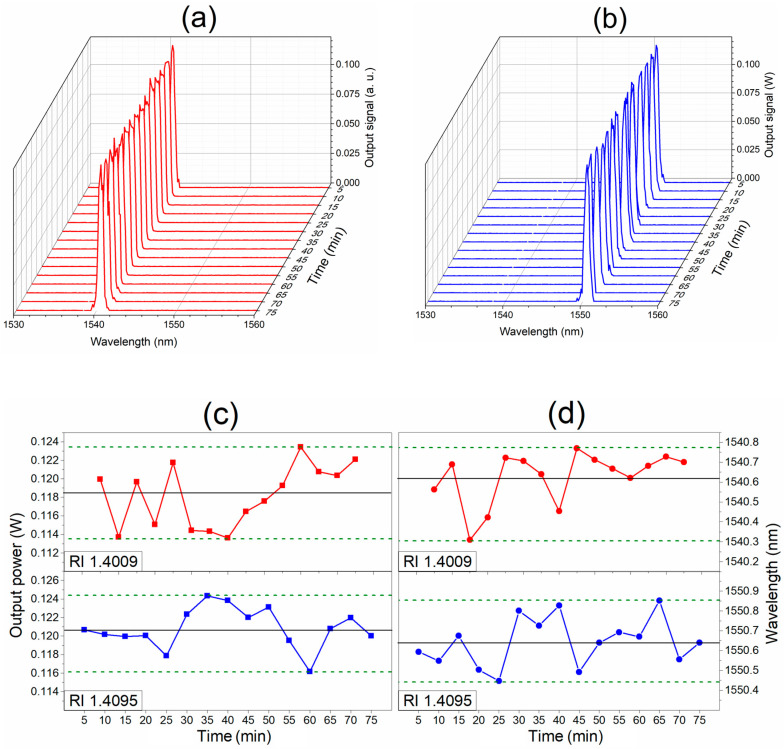
Stability characterization of the RI fiber laser sensor: (**a**) set of output spectra for RI of 1.4009; (**b**) set of output spectra for RI of 1.4095; (**c**) plot of the peak power stability; and (**d**) plot of the stability for the wavelength of the laser line.

**Table 1 sensors-23-03045-t001:** Comparison in terms of Q-factor of some reported fiber-based refractometers and the proposed approach.

Ref.	Sensitivity (nm/RIU)	FWHM (nm)	Visibility (dB)	Q Value	RI Range	Structure	Notes
[31]	259.85	~3.7	35	6.39 × 10^5^	1.333–1.381	SMS	
[32]	286.2	~1.1	16	1.19 × 10^6^	1.33–1.45	SMF-Etched SMS	
[33]	261.9	~0.3	18	4.12 × 10^6^	1.3333–1.3737	Multi-tapered SMS	
[34]	188	~2.8	27	3.41 × 10^5^	1.33–1.40	Mach-Zehnder SMSMS interferometer	
[35]	580.269	~0.9	17	6.63 × 10^6^	1.4–1.45	Cascaded NCF with long period fiber grating	
[36]	1130	~3	15.5	6.60 × 10^6^	1.3333–1.3434	SMF Fabry-Perot interferometer tip	
[37]	1214.7	~12	17	2.09 × 10^6^	1.3678–1.4009	Lossy mode resonance AZO-coated NCF	
[38]	2848	~70	3.2 (48% transmission)	3.71 × 10^5^	1.3328–1.3853	Surface plasmon resonance silver-coated NCF	Visible wavelength range
[23]	−2953.44	0.0164	42	2.23 × 10^10^	1.33302–1.33402	Core-offset Mach-Zehnder interferometer	Laser sensor
This work	1130.28	0.018	45.2	3.21 × 10^9^	1.3939–1.4237	Fiber ball lens interferometer	Laser sensor

## Data Availability

Data are unavailable due to privacy or ethical restrictions.

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
