# Peer review of "Refractive Index Fiber Laser Sensor by Using a Fiber Ball Lens Interferometer with Adjustable Free Spectral Range"

_sensors, 2023, doi:10.3390/s23063045_

Round 1

Reviewer 1 Report

1.  In last paragraph under introduction section, authors need to explain what is the difference between their proposed technique with the existing technique. Also need to add novelty statement of their works.

2.      In line 64, how is the ball diameter of 300 um been selected for this work?

3.      How are the stability performances of the proposed sensor? At least tested in one or two RI level.

4.      In order to ensure practicality of the proposed sensor, how is the reproducibility and repeatability of the proposed sensor?

5.      Need to describe how much the proposed method improved as compared to the conventional technique (without FBL) at least in term of sensitivity.

Author Response

  1. In last paragraph under introduction section, authors need to explain what is the difference between their proposed technique with the existing technique. Also need to add novelty statement of their works.

Authors’ response

Thank you for the time you expended reviewing or work. We really appreciate your comments and suggestions. At the introduction section an explanation of the difference between our work (laser sensor) and the existing techniques (broadband sensors) was included. In this regard, the novelty is that the proposed sensor using a FBL interferometer within a laser cavity significantly improves the refractive index detection compared with reported sensors using the FBL interferometer for detection directly from the interference spectrum. The determination of the wavelength displacement as optical interrogation is more precise and straightforward to determine from a laser line than from a broad-band spectrum. This novelty statement was also included at the introduction section of the revised manuscript.

  1. In line 64, how is the ball diameter of 300 um been selected for this work?

Authors’ response

The FBL was fabricated by using a dedicated program of a specialized fiber fusion splicer. In this case, the fabrication parameters were set to diameter of 300 um since for diameters up to this value the fabricated FBL by this device tend to increase their ellipticity and lose center symmetry with the fiber body (because of the ball weight). Then, in accordance with the results obtained by Li et al. at Ref. [21], we select a 300 um diameter to almost reach the optimized diameters reported for MMF-based FBL where, as the authors state, that large-diameter microspheres are more suitable to be used as follow-up experimental objects, where the diameter of the fiber ball ranging from 330 to 360 µm is recommended.

This information was added at the discussion of Figure 1, section “fiber ball lens interferometer” of the revised manuscript.

  1. How are the stability performances of the proposed sensor? At least tested in one or two RI level.

Authors’ response

Thank you for the comment. In the revised version of the manuscript a new set of figures (Figure 9) was included to demonstrate the stability of the laser line for the proposed sensor. A couple of set for laser spectra in long term measurements were added for two different refractive index levels. For each set of measurements, the analysis for output power and wavelength stability of the laser line is included. In this regard, since the proposed sensor is based on wavelength displacement interrogation, the stability of the peak power is not so relevant compared to the wavelength stability of the central wavelength. However, the obtained results demonstrate adequate stability results for both variables. In case of the wavelength stability, the variations are in a range of ~0.4 nm but the averaged wavelength is ~0.02 nm away from the linear fit values. The peak power stability shows low variations, less than 7% from the average peak power.

  1. In order to ensure practicality of the proposed sensor, how is the reproducibility and repeatability of the proposed sensor?

Authors’ response

The repeatability of the proposed sensor has been proved due to the results shown in the manuscript were obtained in different sessions, implementation conditions, without controlled environments. At room temperature and without significant light isolation. In order to demonstrate the repeatability of the sensor, for the revised manuscript was included a discussion for 15 sequential measurements from the lower to the higher RI levels. The results were added as Figure 8 and its corresponding discussion.

Also, as it is shown in the recently obtained results for stability in Figure 9 where the measurements are consistent with the previous ones obtained, in which the error in the interrogation does not compromise the prediction given by the sensitivity model, obtained in the linear fit.

Regarding the reproducibility of the sensor, the FBL structure is fabricated by using a specialized and dedicated fiber splicer. The fabrication process is controlled and monitored with high precision, then, the reproducibility of the FBL fabrication is not compromised.

  1. Need to describe how much the proposed method improved as compared to the conventional technique (without FBL) at least in term of sensitivity.

Authors’ response

Thank you for the comment. In the revised version of the manuscript the Table 1. Here, the sensitivity, FWHM and the visibility were compared between some reported fiber refractometers with different fiber structures and the proposed approach, through the calculation of the quality factor Q, for each work. Also, the RI range for each reported work is included in the Table. A discussion about the results of the comparison was included in the revised manuscript.

Reviewer 2 Report

In this paper, the authors present a fiber laser refractometer based on a fiber ball lens interferometer, the linear cavity erbium-doped fiber laser uses an FBL structure acting as a spectral filter and sensing element for determining the RI of a liquid medium surrounding the fiber. The optical interrogation of the sensor is the wavelength displacement of the generated laser line as a function of the RI variations. The results indicate that the wavelength of the generated laser line is a linear function of the RI variations on the medium surrounding the FBL with a sensitivity of 1130.28 17 nm/RIU. This article is clear, concise, and suitable for the scope of the journal. Several suggestions are supplied:
1. Suggest the authors supply more detail about the repeat performance of the device.
2. Suggest the authors supply more detail in the sentence about the Beam trace simulation of the intensity distribution.
3. Suggest the authors supply more detail in the sentence about the residual error between the central wavelength of the laser line and the 194 linear fitting value for each RI variation.
4. Suggest the authors check small typos if have any.
5. The authors present here may be used for marine environment monitoring, suggest the authors enhance the introduction part with one last review on this topic:
Optical fiber sensing for marine environment and marine structural health monitoring: A review Optics and Laser Technology, 2021.

Author Response

In this paper, the authors present a fiber laser refractometer based on a fiber ball lens interferometer, the linear cavity erbium-doped fiber laser uses an FBL structure acting as a spectral filter and sensing element for determining the RI of a liquid medium surrounding the fiber. The optical interrogation of the sensor is the wavelength displacement of the generated laser line as a function of the RI variations. The results indicate that the wavelength of the generated laser line is a linear function of the RI variations on the medium surrounding the FBL with a sensitivity of 1130.28 17 nm/RIU. This article is clear, concise, and suitable for the scope of the journal. Several suggestions are supplied:

  1. Suggest the authors supply more detail about the repeat performance of the device.

Authors’ response

We really appreciate all your comments and suggestions. In the revised version of the manuscript are included results for new measurements to demonstrate the repeatability of the proposed laser sensor. In this regard, 15 sequential measurements from the lower to the higher RI of the liquid measured were carried out. From the results Figure 8 and its discussion was included to demonstrate the repeatability of the sensor.

Also, results for the stability of the sensor were included in Figure 9. The results contribute to demonstrate the repeatability of the sensor, since the measurements are consistent with the previously obtained, in which the error in the interrogation does not compromise the prediction given by the sensitivity model, obtained in the linear fit.

  1. Suggest the authors supply more detail in the sentence about the Beam trace simulation of the intensity distribution.

Authors’ response

Thank you for the suggestion. In the revised version of the manuscript more detailed discussion of the beam tracing simulation was included. Also, the coordinate axes of the graphs were modified to facilitate the view of the simulated results.

  1. Suggest the authors supply more detail in the sentence about the residual error between the central wavelength of the laser line and the 194 linear fitting value for each RI variation.

Authors’ response

We appreciate the comment. For the revised manuscript, the residual error graph was removed in order to demonstrate a more detailed analysis for the repeatability of the sensor. In this sense, the new information includes 15 sequential measurements from lower to higher refractive index, with which the repeatability of the sensor is analyzed. The results of the average absolute error are shown in Figure 8 and its corresponding discussion. In the case of residual error results for a measurement, the results are included in the error bars in Figure 7a and in the revised version of the manuscript the discussion of residual error was integrated into the discussion of that figure.

  1. Suggest the authors check small typos if have any.

Authors’ response

The new version of the manuscript was revised in order to amend typos and ensure grammar corrections.

  1. The authors present here may be used for marine environment monitoring, suggest the authors enhance the introduction part with one last review on this topic:

Optical fiber sensing for marine environment and marine structural health monitoring: A review Optics and Laser Technology, 2021.

Authors’ response

Thank you for the suggestion. In the revised version of the manuscript the suggested reference was included to enhance the introduction section, regarding applications of fiber refractometers.

Reviewer 3 Report

In this manuscript, the authors proposed a refractive index fiber laser sensor by using a fiber ball lens interferometer with adjustable free spectral range. The theoretical analysis and experimental results show that the proposed fiber laser sensor is feasible. However, there are some points should be emphasized and interpreted.

1.What parameters are related to the sensitivity of the designed sensor? Why choose the parameters given in the manuscript? It is suggested to provide additional explanations.

2. What is the temperature characteristic of the sensor? It is suggested to add supplementary explanation of the experiment

3. How to control the ball diameter and shape? How about the consistency of manual operation? It is recommended to add sensing samples.

4. The coordinates of the figures are not clear, such as figure 3. It is recommended to modify them.

5. What is the resolution of refractive index detection?

6. How about the repeatability of refractive index detection?

7. Please comment on the sensor resolution to refractive index.

8.Authors should carefully check the manuscript before submission.

Author Response

In this manuscript, the authors proposed a refractive index fiber laser sensor by using a fiber ball lens interferometer with adjustable free spectral range. The theoretical analysis and experimental results show that the proposed fiber laser sensor is feasible. However, there are some points should be emphasized and interpreted.

1.What parameters are related to the sensitivity of the designed sensor? Why choose the parameters given in the manuscript? It is suggested to provide additional explanations.

Authors’ response

Thank you for all the comments and suggestion, we really appreciate them. The proposed FBL was designed in accordance with the results obtained by Li et al. at Ref. [27]. In this regard, we selected a 300 um diameter to almost reach the optimized diameters reported for MMF-based FBL where, as the authors state, that large-diameter microspheres are more suitable to be used as follow-up experimental objects, where the diameter of the fiber ball ranging from 330 to 360 µm is recommended. This information was added at the discussion of Figure 1, section “fiber ball lens interferometer” of the revised manuscript. The diameter of the ball determines the reflectivity of the FBLI since the FBL acts a coupling element for light propagated back and forth through the ball lens. Then, the intensity of the interfered light coupled back to the fiber defines the visibility of the interference spectrum. Part of this clarification was included in the beam trace results.

  1. What is the temperature characteristic of the sensor? It is suggested to add supplementary explanation of the experiment

Authors’ response

Thank you for the comment. The temperature response of the fiber ball lens was recently studied by Zhang et al. in Ref [21]. For the reported work the FBL was studied in order to minimize (near to zero) the crosstalk temperature in a proposed liquid level sensor. In that case, the sensitivity to temperature of the FBL, around 40 pm/°C, compromises the liquid level measurements since slight wavelength shift are obtained for both variables. In our case, the low sensitivity to temperature of the FBL structure can be neglected from high sensitivity to refractive index measurements of 1130.28 nm/RIU. Part of the explanation was added at the results and discussion section.

  1. How to control the ball diameter and shape? How about the consistency of manual operation? It is recommended to add sensing samples.

Authors’ response

Regarding manual operation for FBL fabrication, several authors (Refs. [27-30]) have been reported their methods and results fabricating the FBL by electric arc discharge from a manual fusion splicer. In their studies the main objective is the reproducibility of the FBL with optimized parameters. The discharge parameters studied are mainly time and electric power to control the ball diameter and shape.

In our case, we are not fabricating the FBL from a manual process. Our FBL was fabricated by using a fusion splicer with special fiber processing features (Fujikura ArcMaster FSM-100M, Tokyo, Japan) and the dedicated ball lens arc fusion program of its software (AFL Fiber Processing Software FPS, ver. 1.2b, Tokyo, Japan). The fabrication process is controlled and monitored with high precision. As a result, the reproducibility of the FBL fabrication is not compromised.

In order to clarify this point, the discussion about the fabrication of the FBL at the section “Fiber ball lens interferometer” was modified including part of this explanation.

  1. The coordinates of the figures are not clear, such as figure 3. It is recommended to modify them.

Authors’ response

Thank you for the suggestion. In the revised version of the manuscript, the coordinates of Figure 3 were modified to clarify this issue.

  1. What is the resolution of refractive index detection?

Authors’ response

In order to solve this question, in the revised version of the manuscript additional measurements for analysis of repeatability of the sensor were included in Figure 8. From the obtained results for repeatability in Figure 8, it was found that the calculated mean absolute error of wavelength displacement to the average wavelength is around the OSA maximal resolution of 0.04 nm. In this regard, the resolution of the refractive index sensor is limited by the resolution of the OSA.

Part of this explanation was included in the discussion of the repeatability for the refractive index sensor in Figure 8.

  1. How about the repeatability of refractive index detection?

Authors’ response

To demonstrate the repeatability of the sensor, for the revised manuscript was included a discussion for 15 sequential measurements from the lower to the higher RI levels. The results were added as Figure 8 and its corresponding discussion.

Also, as it is shown in the recently obtained results for stability in Figure 9 where the measurements are consistent with the previous ones obtained, in which the error in the interrogation does not compromise the prediction given by the sensitivity model, obtained in the linear fit.

  1. Please comment on the sensor resolution to refractive index.

Authors’ response

In the revised version of the manuscript additional measurements for analysis of repeatability of the sensor were included in Figure 8. From the obtained results for repeatability in Figure 8, it was found that the calculated mean absolute error of wavelength displacement to the average wavelength is around the OSA maximal resolution of 0.04 nm. In this regard, the resolution of the refractive index sensor is limited by the resolution of the OSA.

Part of this explanation was included in the discussion of the repeatability for the refractive index sensor in Figure 8.

8.Authors should carefully check the manuscript before submission.

Authors’ response

Thank you for the comment. Carefully check of the revised version of the manuscript was done. The revised version includes the reviewers’ comments and suggestions as fundamental guidelines in order to clarify the concerns. Additionally, the manuscript was revised in order to amend typos and ensure grammar corrections.

Round 2

Reviewer 3 Report

The manuscript presented a refractive index fiber laser sensor by using a fiber ball lens interferometer with adjustable free spectral range.In this new version, the manuscript was improved, and the authors addressed the comments.